# Aquaporins Respond to Chilling in the Phloem by Altering Protein and mRNA Expression

**DOI:** 10.3390/cells8030202

**Published:** 2019-02-27

**Authors:** Ryan Stanfield, Joan Laur

**Affiliations:** 1Department of Renewable Resources, University of Alberta, Edmonton, AB T6G 2R3, Canada; 2Institut de Recherche en Biologie Végétale, Université de Montréal, Montréal, QC H3T 1J4, Canada; joan.laur@umontreal.ca

**Keywords:** phloem transport, cold-induced inhibition, aquaporins, PIP1, PIP2, sieve tubes, qrtPCR, immunohistochemistry

## Abstract

Previous experiments using heat exchangers (liquid cooled blocks) to chill a portion of plant stem have shown a transient stoppage in phloem translocation and an increase in measured phloem pressure. Although a chilled-induced stoppage of phloem transport has been known for over 100 years, the mechanism of this phenomenon is still poorly understood. Recently, work has highlighted that aquaporins occur within the plasma membrane of the sieve tubes along the entire source-to-sink pathway, and that isoforms of these water channel proteins may change dynamically. Aquaporins show regulatory roles in controlling tissue and cellular water status in response to environmental hardships. Thus, we tested if protein localization and mRNA transcript abundance changes occur in response to chilling in balsam poplar (*Populus balsamifera*) using immunohistochemistry and qrtPCR. The results of the immunolocalization experiments show that the labeling intensity of the sieve elements treated for only 2 min of chill time significantly increased for PIP2. After 10 min of chilling, this signal declined significantly to lower than that of the pre-chilled sieve elements. Overall, the abundance of mRNA transcript increased for the tested PIP2s following cold application. We discuss the implication that aquaporins are responsible for the alleviation of sieve tube pressure and the resumption of flow following a cold-induced blockage event.

## 1. Introduction

The phloem vascular tissue is the predominant passageway for photosynthetically derived nutrients to be propagated around the body of the plant. Within the phloem lies the sieve tube conduit responsible for the transport of not only sugars, but also for the transmission of signals in the form of mRNA [1], amino acids [2] or electrical action potentials [3]. Thus, the phloem is the critical energy transmission pipeline needed for overall plant homeostasis, as well as the coordination of defenses in such events as insect attack [4], viral outbreak [5], and drought stress [6]. Although we are gaining a better understanding about both the control and motive force behind fluid movement within phloem sieve tubes, many questions remain unresolved in terms of its mechanism of action. Previous research on sugar translocation is often focused on loading near source tissue (e.g., leaves) [7,8,9] or unloading near sink tissue (e.g., roots) [10]. However, the intervening transport phloem which connects source and sink is often neglected due to the difficulty of accessing this deeply embedded tissue. The importance of the transport phloem for influencing long-distance translocation cannot be understated, as it may act as an exchange point for water and nutrients [11], which has been modeled to significantly influence pressure profiles [12].

An important set of experiments that have been performed on transport phloem has used the application of a heat exchanger, or cold-block, to rapidly cool a section of the stem. As early as 1912, experiments applying cold to a small section of the stem have been performed and they demonstrated an inhibitory effect on phloem transport [13]. Indeed, the inhibition of translocation via stem cooling has been measured via carbon isotope tracing of the phloem in a variety of species [14]. While testing 86 species of angiosperms, all dicots and 30% of monocots experienced a cold-induced reduction of translocation. Upon cooling a 10 mm section of the stem, chilling sensitive plants showed an immediate halt, but then recovered within 3–5 min of rewarming; previously, morning glory was also shown to recover after warming within seconds [15]. A more recent study showed that in cow thistle, not only does translocation stop, but a pressure builds up in the sieve tube upstream of the cold-block [16]. This pressure begins to decline to pre-chill levels within 10 min of chilling. The reversibility of the cold response while the cold treatment is still being applied warrants many hypotheses as to the cause of temporary cold-induced phloem blockage.

A continuous sieve tube is made up of the sieve element cells, partitioned end-to-end by sieve plates. One mechanism of cold-induced blockage is that the sieve plates become blocked after chilling due to the dispersion of p-protein filaments which clog sieve plate pores [17]. The dispersion of these so-called forisome p-proteins has been demonstrated to expand rapidly following cooling due to a depolarization of the sieve element membrane in bean plants [18]. However, this mechanism of sieve tube blockage does not fully explain why translocation stops in species which do not possess dispersive p-proteins that are commonly found in legumes [14] or in poplar [19]. In addition, forisomes in *Arabidopsis* that appeared to cover sieve plates did not seem to inhibit phloem transport according to in vivo imaging [20]. An alternative explanation is that the plasma membrane of the sieve elements is somehow disrupted due to chilling [14,15]. This, in turn, may hinder the ability of the sieve element membrane to retrieve assimilates and water that passively leak out along the transport pathway. Since the retrieval of water and solutes is hypothesized to be essential in maintaining mass flow [21], it is plausible that a cold-induced disruption of the plasma membrane may impact flow. In addition, it is hypothesized that either aquaporins or solute transporters are specifically disrupted by cold [16]. 

Aquaporins are intrinsic membrane-bound proteins which are primarily responsible for the passage of water across the plasmalemma or tonoplast [22,23]. A variety of protein isoforms exist [24,25], playing a role in transporting not only water, but also CO_2_ [26] and O_2_ [27]. They occur in a variety of sub-types (isoforms), including plasma membrane intrinsic proteins (PIPs) which are the major water-transporting isoforms found in plants [24]. In response to environmental stress such as cold, aquaporins may react in multiple ways to counteract the loss of the hydraulic conductivity of the tissue that is chilled [28]. Cold has the impact of reducing aquaporin mRNA transcript levels, while simultaneously increasing its protein abundance [29]. In addition, aquaporins are more likely to be phosphorylated when exposed to a chilling event, which is a gating mechanism used to open the water channel [30]. Aquaporins play an important role in mediating the hydraulic conductivity of roots in poplar [31], and likewise respond to chilling through altered transcript abundance [32]. However, it is important to note that both the mRNA expression and protein expression of aquaporins in response to cold may depend on the chilling tolerance of the species tested, as well as the duration of the chilling treatment [33,34].

Previous cold-block experiments on phloem transport have mainly focused on translocation rates using isotope tracing [14,15,35] or pressure [16]. However, no studies to date have shown the effect of aquaporin cold response within the sieve tubes. Despite work that shows how aquaporin cellular location, protein and mRNA transcript abundance change in accordance to environmental stress such as cold [36], it is unknown how these parameters change within phloem sieve tubes. Previous work on aquaporins shows that a different pattern of localization occurs between the PIP1 and PIP2 isoforms [37]. This work showed that PIP1s are predominantly found within internal compartments, whereas PIP2s are found to occur mainly in the plasma membrane of the sieve tubes in poplar. Although no known role has been described for PIPs occurring within internal compartments, the endomembrane likely serves as a reservoir for containing PIPs until they are needed at the plasma membrane [38,39,40]. This shows that aquaporins on the plasma membrane may dynamically regulate their abundance within the sieve tubes to compensate for changes in water potential. Thus, the first objective of this study was to ascertain if localization patterns and protein abundance changed in accordance to cold-block treatment using immunohistochemistry. Next, we sought to determine mRNA transcript abundance using reverse transcription polymerase chain reaction (qrtPCR) and how this was changed according to cold-block treatment. According to the work of [16], there was a transient increase in sieve tube pressure immediately after cold-block treatment began in cow thistle. In this previous work, after 2 min of chilling, the sieve tube pressure began recovering to pre-chill levels. The location of these previously studied effects was upstream (towards the photosynthetic source) of the cold-block. Thus, we hypothesized that aquaporins increase in protein labeling abundance at the cold-block site as well as mobilize in greater quantities in the plasma membrane to release water quickly from the sieve tubes following a chilling event. In addition, we expected mRNA transcript abundance to increase as well from stem tissue located around the site of the cold-block application. We found that mRNA transcript abundance increased for certain isoforms of PIP1 and PIP2. Meanwhile, immunohistochemistry showed that the PIP2 signal increased within 2 min in the plasma membrane of sieve tubes, but then decreased after 10 min of cold application. We discuss the implication that aquaporins are acting to regulate the sieve tube pipeline following a cold disturbance. 

## 2. Materials and Methods

### 2.1. Plant Materials 

Initial dormant balsam poplar (*Populus balsamifera* L.) cuttings were taken from the river valley, adjacent to the University of Alberta, Edmonton, Canada (53°31′45.06′′ N, 113°31′2.88′′ W) on 29 March 2017. Cuttings of 10-cm length were prepared as described by [41]. Cuttings originated from the separate branches of trees connected to the same root stock (thus, the genetic origin was from a single parent plant). Briefly, cuttings were soaked in tap water for two days (water was replaced between each day). Cuttings were then transferred to an equal part perlite, vermiculite, Sunshine soil mix #4 (Sun Gro Horticulture, Agawam, MA, USA). Plants were allowed to break bud within a growth chamber set at 18–21 °C and a 16 h photoperiod for 54 days before being transferred to a greenhouse (18–30 °C) for the rest of the growing season. On 31 August 2017, plants of approximately 1.3-m height were transferred to an outside growing area for their overwintering dormancy period. On 15 January, 2018, these overwintered plants were re-potted in fresh Sunshine Soil Mix #4 and transferred to a temperature-controlled growth chamber with 19 °C and 21 °C night/day temperatures, respectively, and a 16 h photoperiod. Photosynthetic photon flux was 363 umol m^−2^s^−1^. Trees were well watered on a daily basis, and a 20-8-8 NPK fertilizer at 200 ppm was applied on a biweekly basis. Experimental sampling began on 26 April 2018. 

### 2.2. Cold-Block Experiment

An aluminum cold-block measuring 10 mm × 10 mm × 13 mm (l × w × h) was fabricated from the University of Alberta Physics Machine Shop to encapsulate a small section of the stem (Figure 1A). The block was divided into two symmetrical pieces so that it could be easily placed and removed from the intact test stem. Each side of the block had 5-mm holes drilled out for the passage of room temperature or cold water. Three blocks in total were designed to accommodate 3-, 4- or 5-mm diameter stems. The block was fastened in the middle of the internode region (Figure 1B) of the stem. Thermal paste (Céramique, Arctic Silver Inc., Visalia, CA, USA) was applied to the stem to allow for good thermal contact between the stem and cold-block. To ascertain the temperature experienced at the cold-block, a thermocouple wire was inserted into the cold-block contacting both the stem and block (blue wire, Figure 1B). Two small screws were used to tighten the block to the stem. Inlet and outlet plastic tubing was used to deliver water to and from the cold-block. A submersible pump (Algreen Products, Cambrige, Ontario, Canada) with a rate of 757 L per hour was used to pump water into the block from a small reservoir of either room temperature or ice water. Temperature values were recorded for the warm and ice water reservoirs as well as cold-block (Table 1). A stop-cock was used to quickly change water flow from room to ice water reservoirs or visa-versa. 

Cold-block measurements were carefully undertaken to minimize the chance that phloem tissue would change because of disturbance. Of crucial importance was the minimization of phloem-induced disturbance via vibration. Even minor shaking may cause the cessation of phloem transport [42]. Thus, any measurements taken in the subsequent experiments were allowed to settle 60 min after the cold-block was fastened to the stem to allow the translocation system to equilibrate to vibrational disturbance, as suggested by [43]. When the cold-block was fastened to the stem, care was taken to not move or shake the plant. To make the plant accustomed to the vibration of water moving through the cold-block, room temperature water was ran through the cold-block for the initial 60 min equilibration time. After this time, cold water was run through the cold-block. As the xylem and phloem are believed to be hydraulically linked, stomatal conductance measurements were taken before and after cold water was run through the block to determine the best sampling times for subsequent experiments.

Stomatal conductance measurements were taken using a SC-1 Porometer (Decagon Devices, Pullman, Washington, DC, USA) for N = 24 test plants. For each plant, conductance measurements were taken every five minutes on the distal leaf located closest to the cold-block (Figure 1B). A total of 12 measurements were taken during the initial 60 min equilibration time (room temperature), after which cold water was run through the block. Another 12 measurements were taken for 55 min after cold-block application. The data for these measurements are summarized (Figure 2). The average stomatal conductance was calculated for each plant prior to cold application. The conductance value for each time sampled after cold application was then subtracted by this average pre-cold conductance value and converted to an absolute value. Note that on average, stomatal conductance was lower post cold-block application (data not shown). These stomatal conductance measurements, in addition to the phloem pressure response times generated from the Gould et al. [16] study, were used to justify sampling times for immunolabeling and mRNA analysis. Our results indicated that deviation of stomatal conductance relative to pretreatment levels peaked after 5 min of cold application. In addition, pressure drop measurements from Gould et al. [16] showed that a phloem translocation stoppage occurred during the first two minutes of the cold-block application. According to our results, after 10 min of the stem cooling, stomatal conductance fell closer to pre-chilled levels, and according to Gould et al. [16], so did pressure. Previous cold-block studies have also shown a recovery of translocation after the stem was rewarmed to pre-chill temperatures. Therefore, four temperature treatments were established for use in immunolabeling and mRNA expression sampling. These treatments were, (1) control treatment: 60 min room temperature water, (2) 2-min chilling: 60 min room temperature followed by 2 min cold water, (3) 10-min chilling: 60 min room temperature followed by 10 min cold water and (4) rewarm: 60 min room temperature, and then 10 min cold, followed by 10 min room temperature water.

### 2.3. Fixative and RNA Stabilizer Solutions

Two solutions were made for preserving the cellular structures for immunolabeling or mRNA analysis. For immunolabeling, the stem samples were kept in chilled Formalin Acetic Acid (FAA). For mRNA analysis, the stem samples were kept in homemade RNA stabilizer solution; the solution was made by adding 117 g of ammonium sulfate and 5.56 mL of 0.75 M sodium citrate to 6.67 mL of 0.5 M EDTA adjusted to a pH of 5.2 in 250 mL of nuclease-free water. 

### 2.4. Sampling for Immunolabeling and mRNA Expression 

Sampling for each tree occurred between 11:00–11:30 each day over an 86-day period between 24 April 2018, and 13 July 2018. The sampling was taken precisely where the cold-block was applied to the stem of the tree (i.e., 13 mm of the stem covered by the block). After the treatment time was completed, the cold-block was carefully and quickly removed. The stem was then quickly cut below the nearest basal leaf, and then the section of the stem with the cold-block was trimmed using a fresh razor blade. This 13-mm stem section was cut in half and dropped into one of two solutions (a) 4 °C chilled FAA for immunolabeling or (b) RNA stabilizer for mRNA expression data. After 30 min, the samples for immunolabeling were transferred into fresh chilled FAA, and once more after 6 h. The samples for mRNA analysis were transferred to a −20 °C freezer until the start of qtPCR. For immunolabeling, a total of 6 controls and 4 plants from each experimental treatment were examined, for a total of 18. 

### 2.5. Sectioning and Immunolabeling

The stem sections were kept in FAA fixative for at least 72 h prior to embedding. Organs were then paraffin embedded in a Leica TP 1020 tissue processor (Leica Microsystems, Wetzlar, Germany) before going into wax block molds. Longitudinal or transverse sections of 7 µm thickness were made on a rotary microtome and transferred directly onto Probe-on Plus (Fisher Scientific, Pittsburgh, Pennsylvania) slides flooded with deionized water. The slides were kept on a hotplate set to 50 °C until the sections flattened out on the water and were then placed in a drying oven for at least 24 h at 37 °C.

Immunolabeling was performed following the methodology of [44], previously described in [37]. Antibodies used to target aquaporins in *Arabidopsis* (*At*PIP1;3) were used and targeted the 42 N-terminal sequence of this protein; this antibody had confirmed reactivity with balsam poplar aquaporin PIP1s according to Western Blot analysis [37]. For PIP2s, antibodies were raised against the conserved 10-sequence C-terminal of this protein (with confirmed reactivity in balsam poplar, as previously demonstrated [37]). Given the conserved nature of our PIP2 antibody, it targeted all PIP2s indiscriminately during the immunolabeling process. 

To summarize the immunolabeling methodology, the slides were dewaxed in Safeclear^®^ Xylene Substitute (Fisher Scientific), rehydrated in an ethanol series and then washed in phosphate buffered saline (PBS). After a step in post-fixative and another wash in PBS, the slides were transferred to blocking solution (BS) and then washed briefly in low-salt water washing buffer (LWB) before primary antibodies were applied. Approximately 80–100 µL of *At*PIP1;3 and PIP2 aquaporin antibodies were applied to the slides. After 16–24 h of incubation at 4 °C in a dark humid container, the slides were washed in LWB and secondary antibodies were applied. Approximately 80–100 µL of Pre-absorbed [1/500] Alexa Fluor (Fisher Scientific) 488 conjugated goat anti-mouse and Alexa Fluor 568 conjugated goat anti-chicken were applied to the slides for 2 h at 37 °C. Secondary antibodies were then removed with LWB, quickly washed in double-distilled H_2_O and mounted in Slow Fade Gold (Fisher Scientific). Cover slips were sealed in using nail polish. 

### 2.6. Microscopy and Image Analysis 

Confocal microscopy was performed on a Zeiss LSM 700 (Carl Zeiss AG, Oberkochen, Germany) operated on Zen Black 2011 edition software. A 63x oil immersion lens was used to capture images. Laser power was set to 5.5% to excite Alexa Fluor 488 (PIP2) and to 2% to excite Alexa Fluor 568 (PIP1). Camera gain was 700 for both color channels and the pinhole diameter was set to 1 Airy unit.

For 3D Structured Illumination Microscopy (3D-SIM), images were captured as previously performed in [37]. Briefly, longitudinal stem sections of 7 µm thickness were adhered to poly-l-lysine coated cover glass (#1.5 thickness). The sections were suspended on a small pool of water above the cover glass and allowed to flatten; heat was applied from a slide warmer below (set to 37 °C) to facilitate adherence of the tissue to the cover glass. Once the cover glass dried overnight, the tissue was processed as detailed as above, with the exception of allowing the secondary antibody to incubate for 4 h at room temperature. Once the tissue was properly labeled with PIP1 and PIP2 antibodies, a drop of Slow Fade Diamond (Fisher Scientific) mounting medium was applied, and the cover glass was sealed in using nail polish to the middle of a standard microscope glass slide. To carry out 3D-SIM microscopy, a Deltavision OMX was used. PIP1 and PIP2 fluorophores were excited using 568 nm and 488 nm laser lines, respectively. Laser power was set to 1%, and the exposure times were set to between 100 and 200 ms.

Image analysis was carried out using Image-Pro Premier Version 9.2 (Media Cybernetics, Rockville, Maryland). Transverse section images from each sampled tree were analyzed to view sieve elements. The sieve elements within each image were manually traced based upon their PIP2 outline. The use of software allowed for three main types of data to be collected for highlighted sieve elements: (1) average signal intensity (luminance µm), (2) internalization of aquaporin signal and (3) sieve element area (µm^2^). Internalization of aquaporins was a categorical value (i.e., sieve element appearing with mostly internalized aquaporin signal) which was determined using the margination and heterogeneity tools of Image Pro (as performed in [37]). Generally, if the sieve elements had margination values of ≤0.56, they were classified as having an internalized aquaporin signal; conversely, if the value was >0.56, their aquaporin signal was classified as membrane-bound.

### 2.7. qrtPCR Analysis 

Gene transcript measurements by quantitative real-time PCR sections of the stem segments corresponding to the cold-block application (~15 mm) were collected and submerged in RNAlater stabilization solution (Ambion, USA) until further processing. The samples were always collected between 10:00 h and 11:30 h to minimize any diurnal effects on aquaporin expression. 

Under a binocular microscope, >40 mg of phloem-enriched tissue was dissected by peeling off the interior layer of epidermal peels using fine forceps. Control tissue containing the remaining epidermis and the xylem tissue was also stored at −80 °C. Total RNA was extracted using the CTAB method [45]. RNA quality was assessed on an agarose gel and quantified with a spectrophotometer (Nanodrop ND-1000, Thermo Scientific, Wilmington, DE, USA). RNA was treated as previously described [46]. Putative stem-expressed AQP genes were selected, PtPIP1;1 (Potri.010G191900), PtPIP1;4 (Potri.006G098100), PtPIP2;4 (Potri.008G039600), PtPIP2;5 (POPTR_0006s12980) and PtPIP2;8 (POPTR_0009s01940) [47], and specific primers (Appendix A) were designed using the QuantPrime online tool [48]. PCR efficiency (E) was determined from a five-point cDNA serial dilution, according to: E = 10[−1/slope]. All selected primer pairs showed correlation coefficients of R2 > 0.98 and primer efficiency values ranging between 1.92 and 2.03, and specificity was checked using melting curves. Real-time qPCR was performed on an Applied Biosystems viiA™ Real-Time PCR system (Applied Biosystems, Foster City, CA, USA). Relative gene expression was measured according to [49], using the 2^−ΔΔ*C*T^ method. The expression values were normalized to the housekeeping gene, *Elongation Factor 1B* (Potri.001G224700; [50,51]). Relative gene expression was determined as the fold change of an AQP isoform at a given condition relative to its expression under control conditions. Real-time PCR was carried out using three biological replicates each with three technical replicates.

### 2.8. Data Analysis 

A one-factor ANOVA was used to determine whether a significant difference existed between the cold-block treatments from the response variables of signal intensity, aquaporin internalization and sieve element area. Multiple comparisons were made for outcomes determined to be significantly different (*p* < 0.05) using the Tukey Test. Sigma Plot Version 13.0 (Systat Software, San Jose, California) was used to compute all statistical tests.

## 3. Results

Visually, immunolabeling of the stem organ sieve elements in cross-sectional view appeared to have different intensities (Figure 3). The overall phloem area can be seen stained with aniline blue (Figure 3A). For antibody labeled specimens, a primary antibody control (no primary antibody applied) showed only background fluorescence using confocal microscopy (Figure 3B). In contrast, the four experimental treatments showed various intensities of PIP1 (red) and PIP2 (green) (Figure 3C–F). The sieve elements could be identified through their labeling of PIP2 within the plasma membrane. Of interest were the moderate aquaporin intensities within the sieve elements of the control and rewarm treatment (Figure 3C,F). In contrast to these, the 2-min chilling treatment (Figure 3D) had very intense PIP2 antibody signals within sieve element plasma membranes whereas the 10-min chilling treatment had markedly diminished signals (Figure 3E). 

The higher resolution longitudinal stem sections viewed using 3D-SIM (Figure 4) also showed a similar pattern to cross-sectional confocal images for sieve element aquaporin labeling. Plasma membrane labeling for the non-chilled control treatment showed high levels of PIP2 labeling of the sieve element membranes (green), with some internal membrane labeling of PIP1 (red) (Figure 4A). After 2 min of chilling, the sieve elements showed strong plasma membrane PIP2 labeling (Figure 4B). The 10-min chilling treatment showed markedly lower labeling response in the plasma membrane (Figure 4C, between arrow points); meanwhile, some labeling was shown to occur within the internal membranes (Figure 4C, asterisk), although the signal was rather weak. After stem segments were rewarmed for 10 min with room temperature water, there was a recovery of the PIP2 signal in the plasma membrane (Figure 4D, between arrow points) and within the internal membranes (Figure 4D, asterisk). 

To ascertain if there was a quantitative difference in antibody labeling between treatments, image analysis was performed on 30 confocal cross section images taken from each sampled tree from within each treatment (N = 18). Quantitative intensities of the PIP2 signal were significantly greater in the 2-min chilling treatment than the control and 10-min chilling treatment (Figure 5A; DF = 17, F = 5.273, *p* < 0.05). Additionally, the 10-min chilling treatment had a 1.6-fold lower antibody signal than the control treatment and was on average 3.1-fold less than the 2-min chilling treatment. In contrast, the rewarm treatment was not significantly different from the other treatments. Comparing the ratio of PIP2:PIP1 signal intensity, there was no significant difference found between treatments (Figure 5B; DF = 17, F = 0.717, *p* ≥ 0.05). In terms of the sieve element cross-sectional area, there was no significant difference found between treatments (Figure 5C; DF = 17, F = 2.362, *p* ≥ 0.05). Although there was a trend towards greater internalization of aquaporins from the plasma membrane to internal membranes in the experimental treatments vs. the control, no significant difference was found (Figure 5D; DF = 17, F = 1.138, *p* ≥ 0.05).

Using qrtPCR analysis, mRNA transcript abundance was assessed between the different cold-block treatments (Figure 6). A total of five PIP aquaporin genes and one sucrose transporter (SUT) gene was measured. The reference transcript abundance among the four treatments was made relative to the control (i.e., control transcript abundance was always 1). Overall, PIP2;4 showed the most significant treatment effect as it significantly increased 3-fold in comparison to controls for the 10-min chilling treatment and also saw a more than a 2.5-fold increase for the rewarming treatment (Figure 6A). Although PIP2;5 did show an increase for the rewarming treatment, this was not statistically significant (Figure 6B). In comparison, PIP2;8 showed a significant 2-fold increase over controls for the rewarming treatment (Figure 6C). In contrast to PIP2s, overall, PIP1s did not increase substantially in response to the cold (Figure 6D,E). However, the PIP1;4 gene did show a significant gain of over 2-fold for the 10-min chilling treatment (Figure 6E). Finally, the sucrose transporter tested, SUT4, showed a significant ~2.5-fold increase for the rewarming treatment in comparison to the control (Figure 6F). 

## 4. Discussion

### 4.1. PIP2 Signal Increases Substantially Following Cold Treatment, Then Declines

The immunolabeling experiments showed a significant increase in the labeling intensity for PIP2 proteins 2 min after cold was applied to the stem. However, after 10 min, the aquaporin labeling intensity dropped significantly to lower than that of the controls. In chilling tolerant maize lines, leaves exposed to chilling temperatures of 12 °C showed an increase in PIP2;3 aquaporin labeling density within the sieve tubes after 28 h of cold exposure relative to controls [34]. In contrast, the chilling sensitive line of maize leaves showed a decline in PIP2;3 density in the sieve tubes relative to controls according to their TEM analysis. In a separate study, it was found that maize roots exposed to 5 °C chilling for 3 days showed a significant increase in PIP2 protein abundance according to immunoblot intensity [29]. In rice roots subjected to a 10 °C chill for one day, *Os*PIP2;5 protein abundance increased by ~30% in comparison to controls, according to SDS-page results [33]. These prior studies show an increase in protein abundance and/or labeling following a long-duration chilling regime. In contrast, the current study shows this increase much more rapidly. This could mean that the aquaporin response to chilling is much quicker than previous researchers have reported. The quick aquaporin response could be, in part, due to a tissue-level response to control hydraulic conductivity. The effect of chilling causes a greater amount of water stress at both the root [33] and leaf level [34,52]. This stress is a result of not only viscosity increases that inhibit root water uptake, but also of the decline in the hydraulic conductivity of the plasma membrane. The loss of hydraulic conductivity due to chilling may also be a result of decreased fluidity of the plasma membrane [53]. Thus, the uptick in PIP2 aquaporin protein signal within the sieve tubes observed in the current study could be to counteract inadequacies of the plasma membrane to transfer water via passive diffusion. 

Apart from a direct response to cold, the aquaporins in the sieve tubes could also be responding to pressure. The sieve tubes experience a transient build-up in pressure following cold-block application [16]. Workers found that after 2 min of the cold-block application, there was a >2-fold pressure increase inside the sieve tubes of cow thistle, and a complete stoppage of translocation. However, translocation began to resume after 8–10 min of cold application and pressure returned to pre-chill levels. Similarly, in the current study we found that aquaporin signal intensity was maximized after only 2 min of cold application, but then fell to lower than that of control plants after only 10 min of chilling. Aquaporins are hypothesized to respond mechanically to high pressure by closing [24,28]. In a sieve tube which is blocked due to localized chilling, the transient increase in pressure may have caused aquaporins to temporarily close. One way the membrane can counteract this loss of conductivity may be from upregulating aquaporin abundance in the plasma membrane. More aquaporin abundance means greater rates of radial water flows into the sieve tubes [12], which have the effect of decreasing viscosity and lowering pressure gradients. In effect, this upregulation of aquaporins in the membrane may be a way to counteract some aquaporin gating via a temporary pressure surge. However, after the pressure gradient is corrected for, the aquaporins are no longer needed and may be downregulated in the plasma membrane (as seen in the present study, 10-min chilling treatment). Future studies are needed to parse out the contribution due to pressure and cold-induced changes in aquaporin expression within the sieve tube plasma membranes.

Thus, we accept our hypothesis that aquaporin protein abundance in the plasma membrane increases in response to a perceived chilling blockage event. However, we must reject our hypothesis that the localization pattern of aquaporins changes in response to cold as we did not find a significant treatment effect of the plasma membrane: internally located aquaporins. In other words, there was no evidence to suggest that aquaporins relocated from the plasma membrane to internal membranes, or vice versa, following cold-block treatment. The rationale behind testing aquaporin distribution is that this is a potential mechanism to regulate aquaporin abundance at the plasma membrane following disturbance. Our current understanding of PIP trafficking is that they may shift location from the plasma membrane into internalized compartments such as the endoplasmic reticulum or vice versa [38,40,54]. Alternatively, PIPs may be diverted into lytic vesicles for degradation following an environmental disturbance [40]. Instead, it was found that labeling intensity, especially in the plasma membrane, changed following cold-block application. If aquaporin redistribution was not responsible for their increased accumulation in the plasma membrane, what was? In other words, what was the mechanism which caused these rapid changes in abundance in the 2-min and 10-min chilling treatments? In response to changes in the environment, aquaporins change their conformational shape to close or open the water channel [30]. Therefore, it may be speculated that this conformational change impacted the epitope of the aquaporin in such a way as to either encourage or interfere with antibody labeling. In the case of the 2-min chilling treatment, perhaps the epitope became more available following a cold-induced conformational change to open the channel. Alternatively, in the case of decreased aquaporin accumulation in the plasma membrane (10-min chilling), aquaporin closing could have made the epitope less available for binding. Alternatively, the proteins may have been removed from the membrane entirely and destroyed in lytic vesicles after prolonged chilling. Both possibilities point to either the enhancement or retardation of aquaporin activity in the plasma membrane following cold application. In summary, we hypothesize that aquaporins may have had their epitopes altered, or may have been quickly degraded following cold-block application.

### 4.2. mRNA Transcript Abundance Changes Depending upon Chilling Treatment

Although the protein expression patterns provide an explanation of how the sieve tube plasma membranes respond to chilling, mRNA analysis portrayed a more complicated picture. Overall in the current study, we found that mRNA for PIP1 isoforms were mostly unaltered upon cold treatment, except for PIP1;4 which showed a significant 2-fold increase after 10 min of chill time. In contrast PIP2s showed a more robust response, in particular with PIP2;4, reaching its maximum 10 min after cold-block application. Thus, we may only partially accept our hypothesis that aquaporin genes are upregulated following a cold event, as only certain aquaporin genes are impacted. This can be relayed to our immunolabeling results for which our antibodies bind to all PIP1 or PIP2 epitopes [37]; we can tentatively conclude that PIP1;4 or PIP2;4 may be the most likely candidates to enhance protein localization at the plasma membrane in the sieve tubes following chilling.

A curious difference between the protein immunolabeling and mRNA transcript abundance was the timing for which the two changed. Whereas the immunolabeling results first showed a strong PIP2 increase after 2-min chilling and a drop-off after 10 min, mRNA transcript abundance took at least 10 min of chilling to respond. However, the aquaporin response at the mRNA transcript and protein expression level need not always correlate. For example, it was shown that while PIP1 and PIP2 protein abundance increased after cold in maize roots, mRNA expression levels for these aquaporins declined [29]. This shows that mRNA transcript and protein abundance fall under complex regulation. In addition, other studies have also shown the timing of transcript and protein abundance following an environmental stress need not sync. Specifically, changes in mRNA expression may be slower than changes at the protein level. In one example in response to salt stress, PIP transcript levels took between 2–4 h to decline, whereas changes in protein abundance occurred within 30 min [39]. The duration of chilling may also impact the protein and mRNA expression profiles of PIP genes differently. In a chilling experiment involving the roots of *Arabidopsis*, it was found that the overexpression of PIP1;4 and PIP2;5 in mutant plants counteracted the root loss of cellular hydraulic conductivity [32]. In particular, expression levels in the roots for the aquaporin gene PIP2;5 only increased significantly after 24 h of total chilling time. In contrast for PIP1;4, there was a significant increase in transcript abundance after only 1 h of chilling. These results indicate that PIP genes respond differently to cold stress. The mechanism behind this could be the differing roles of PIP1 and PIP2 aquaporins. 

The roles of PIP1 and PIP2 aquaporins may be functionally different. Whereas PIP2 serves primarily as a water channel [39], PIP1 may serve to transport other molecules such as CO_2_ [26] or O_2_ [27]. A preponderance of studies (see [39] for a review) show that PIP1 has a much lower ability to transport water than PIP2, and that localization studies often find PIP1 within the internal membranes rather than the plasma membrane. This finding of internal PIP1 localization has been documented in the sieve tubes [37] and points to the idea that PIP1 has a regulatory role rather than a primary water transport role in plant plasma membranes. One regulatory mechanism that PIP1 may possess is the ability to modulate the activity of PIP2 as a water channel. Previous results indicate that PIP isoforms show synergistic relationships with one another. Differing ratios of PIP1 and PIP2 subunits within heterotetramers have a contrasting effect on the membrane permeability of measured oocytes [54]. For example, *Zm*PIP1;2 and *Zm*PIP2;5 increased membrane permeability by 2-fold in comparison to when *Zm*PIP2;5 was inserted into *Xenopus* oocytes by itself. This enhancement of the water-transporting capacity of PIP2;5 was only maintained when an excess of PIP1;2 was produced in the oocyte while the addition of PIP1;1 did not produce a synergistic effect. 

Thus it is possible that only certain combinations of PIP1 and PIP2 isoforms garner a synergistic effect, which may change how transcript abundance data are interpreted in response to an abiotic stress event. In the current study, we may speculate that PIP1;4 increased significantly after chilling to bolster the effects of the water-transporting aquaporin PIP2;4. However, this effect has yet to be tested. As has been suggested [39], it is important to test many combinations of PIP1 and PIP2 genes for their potential synergies. In addition, protein level modifications as well as cellular localization may play an even more important role in determining the response of aquaporins to environmental stress. Future studies will need to verify how different ratios of aquaporin transcript abundance (e.g., PIP1 vs. PIP2) affect the control of cellular water permeability in stress conditions. 

## 5. Conclusions

In the current study, we found there to be an increased signal of PIP2 aquaporins in the sieve tube membranes after 2 min of chilling. This response mirrors what has been found physiologically when a stem segment is subjected to cold which causes the sieve tubes to experience a transient increase in pressure and loss of translocation, presumably due to a blockage event. The results of this study provide a mechanism for pressure release following cold application. Potentially, the upregulation of PIP2 abundance in the plasma membrane of the sieve tubes acts as a pressure release valve following cold-block application. Once flow is resumed, aquaporin abundance is then quickly adjusted in the plasma membrane to maintain an adequate pressure profile in the sieve tube. We also found that while the upregulation of mRNA gene expression does not match up with the timing of protein signal change within the sieve tubes, the two need not necessarily be linked. It is likely to be protein-level modification which acts to more immediately rectify disturbances in cellular water balance. Later, altered mRNA transcript abundance could then impact protein-level accumulation in the plasma membranes of affected cells, as needed. This relationship between mRNA transcript abundance and protein expression following cold application will need to be tested in future studies. Although it is still not clear what causes the sieve tubes to become blocked following cold application, the plasma membrane likely has a role in regulating the release and retrieval of both water and sugar following a disturbance to maintain proper pressure gradients. 

## Figures and Tables

**Figure 1 cells-08-00202-f001:**
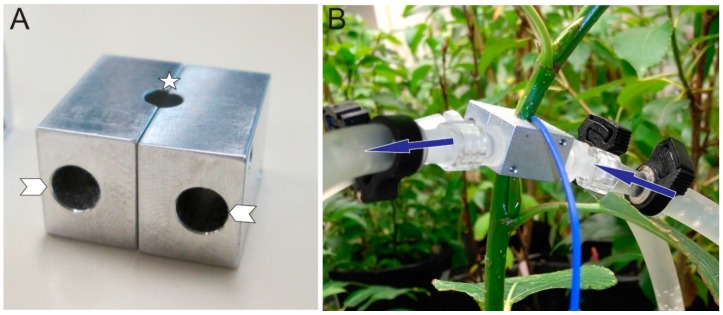
Aluminum cold-block used in experiments. (**A**) The block was designed in two symmetrical pieces for easy removal from plant stems of diameters 3-, 4-, or 5-mm (location of stem insertion, star). Two holes on each subunit were connected to plastic tubing to allow for either room temperature or chilled water to pass through (arrowheads). (**B**) The cold-block was inserted between the internode regions on the stem of the test plant. A thermocouple wire was inserted between the cold-block and stem to assess the temperature of the cold-block which was in direct contact with the stem (blue wire). The plastic tubing would deliver water to each subunit of the block (blue arrows, direction of water flow).

**Figure 2 cells-08-00202-f002:**
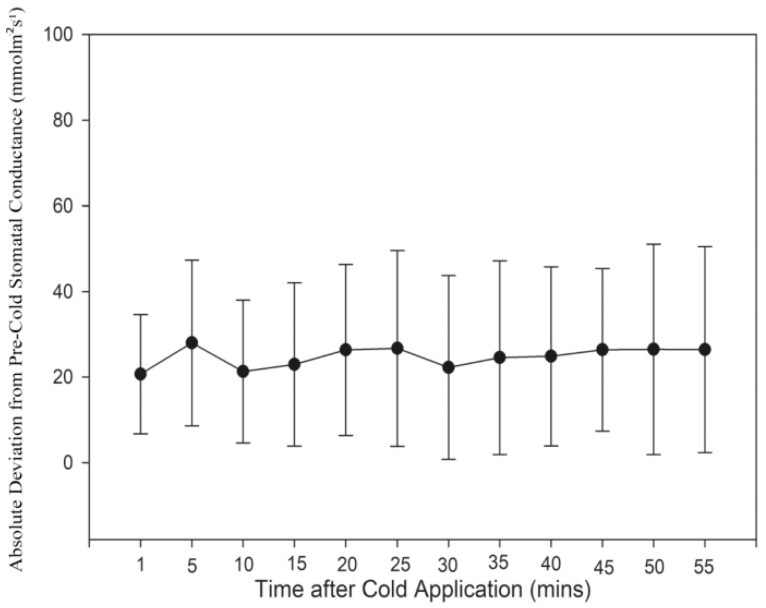
Stomatal conductance measurements after application of cold-block. Each point within the series represents the absolute difference between the average pre-chilled and post-chilled stomatal conductance at each time. The time of 5 min after cold application was observed to have the largest absolute deviation of stomatal conductance from pre-cold stomatal conductance. Bars represent Standard Deviation (SD). N = 24 plants.

**Figure 3 cells-08-00202-f003:**
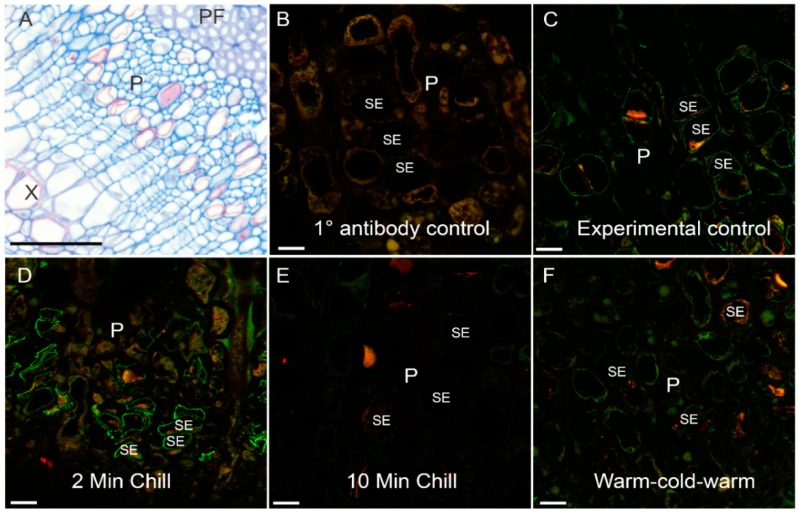
Representative phloem bright field and confocal micrographs showing PIP1 and PIP2 aquaporin labeling following cold-block treatments. (**A**) Cross section of stem tissue showing the phloem region (P) in-between phloem fibers (PF) and xylem (X). Phloem is stained using aniline blue. (**B**–**F**) Confocal laser scanning micrographs of phloem tissue from various cold-block experimental treatments. Red channel = PIP1 labeling, green channel = PIP2 labeling. (**B**) Background fluorescence of the 1° antibody control. (**C**) Experimental control treatment (no cold application) showing moderate labeling of sieve element membranes. (**D**) Two-minute cold application showing enhanced labeling of sieve element membranes. (**E**) Ten-minute cold-block application showing diminished labeling of the sieve element membrane. (**F**) Ten-minute application of room temperature water following 10 min of cold application and finally 10 min of room temperature water; labeling of membranes in this treatment increased slightly over that of the 10-min chilling treatment. Scale bars: (**A**) 60 µm, (**B**–**F**) 10 µm.

**Figure 4 cells-08-00202-f004:**
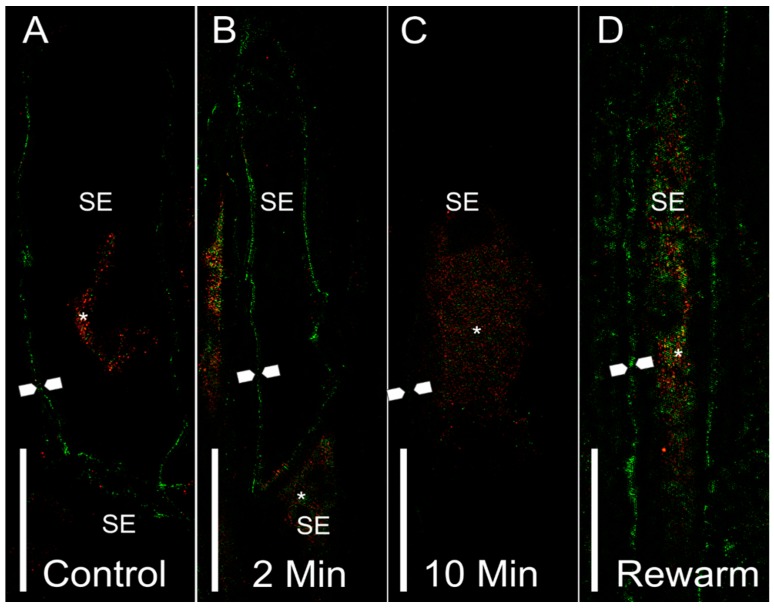
Representative 3D-SIM longitudinal stem sections of sieve elements with PIP1 (red) or PIP2 (green) labeling. (**A**) Sieve elements from a control stem showed PIP2 labeling at the plasma membrane (between arrow points). Some labeling was shown to occur within internal compartments (asterisk). (**B**) Sieve elements from a 2-min chilled stem showed an abundance of PIP2 labeling in the plasma membrane (between arrow points). Some PIP1 labeling was shown in the connecting sieve element (asterisk), as well as a neighboring phloem parenchyma cell. (**C**) A sieve element from a 10-min chilled stem showed an overall reduction of labeling to occur at the plasma membrane (approximate location of membrane shown between arrow points). Some internalization can be seen for PIP1 within internal membrane compartments of the sieve element. (**D**) A sieve element from the rewarmed treatment showed a recovery of the PIP2 signal in the plasma membrane (between arrow points), as well as the PIP1 and PIP2 signals within internal sieve element membranes. Symbols: SE = sieve element, arrow points = location of plasma membrane, asterisk = internal localization of aquaporin signal. Scale bar = 5 µm.

**Figure 5 cells-08-00202-f005:**
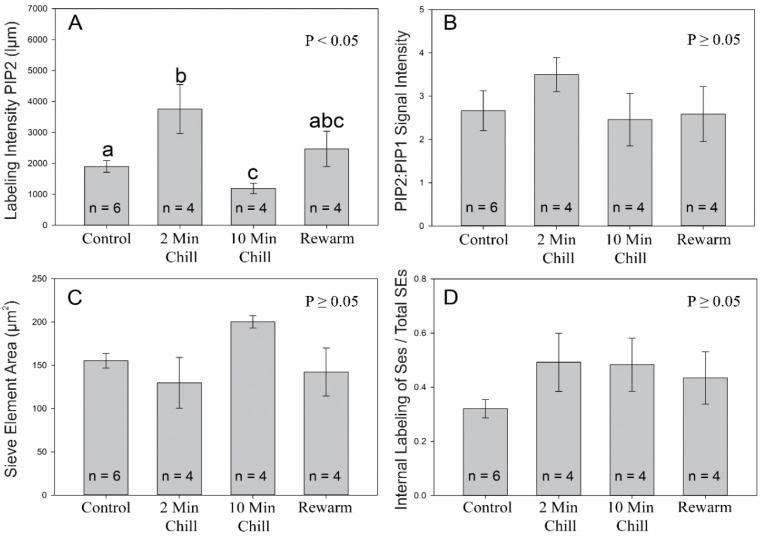
Image analysis of aquaporin labeling from the four experimental cold-block treatments. (**A**) A measure of the pixel signal intensity (luminance micrometers) using the mean intensity value tool in Image Pro software was used for highlighted sieve elements labeled with PIP2 aquaporins. The 2-min chilling treatment had significantly greater intensity values than the control and 10-min treatments (1-way ANOVA). Different letters indicate a significant difference (*p* < 0.05) between treatments (Tukey-Test). (**B**) Ratio of PIP2:PIP1 intensity values of highlighted sieve elements; no significant differences were found between treatments (*p* ≥ 0.05). (**C**) Mean area of individual sieve elements; no significant difference was found between treatments regarding sieve element area (*p* ≥ 0.05). (**D**) The proportion of sieve elements, on average, which were categorized as having an overall internalization of PIP1 and PIP2 aquaporins; no significant difference was found between treatments (*p* ≥ 0.05). Values shown are means ± standard error (SE).

**Figure 6 cells-08-00202-f006:**
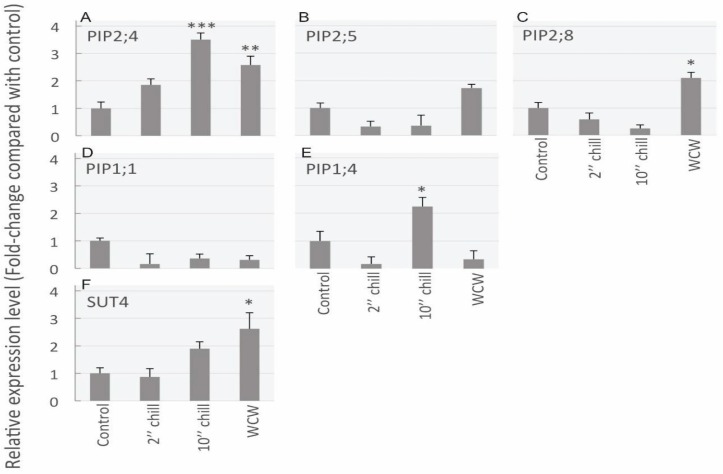
Results from qrtPCR mRNA transcript abundance analysis for five aquaporin genes and one sucrose transporter gene in stem samples from the following four cold-block treatments: control, 2-min chilling, 10-min chilling and rewarm after chilling (WCW). (**A**) PIP2;4 transcript significantly increased after 10-min chilling and rewarming treatments in comparison to control treatments. (**B**) PIP2;5 did not significantly change between treatments. (**C**) For PIP2;8, the rewarming treatment had significantly greater transcript abundance over the controls. (**D**) PIP1;1 did not show significantly different transcript abundance between treatments. (**E**) For the PIP1;4 isoform, transcript abundance declined for the 2-min chilling treatment, whereas it significantly increased for the 10-min chilling treatment. (**F**) The sucrose transporter, SUT4, was shown to have significantly higher transcript abundance for the rewarming treatment in comparison to the control. Asterisks denote significant differences in expression level compared to control levels (one-way ANOVA, followed by Tukey HSD post-hoc test, * *p* ≤ 0.05; ** *p* ≤ 0.01; *** *p* ≤ 0.001). Data are means + standard error (SE) of three biological replicates.

**Table 1 cells-08-00202-t001:** Overall mean temperatures ± standard error (SE) recorded for cold-block experiments. Averages were obtained from stomatal conductance, mRNA and immunohistochemistry experiment collections. N = 1242.

	Room Temp. Reservoir (°C)	Ice Water Reservoir (°C)	Room Temp. Block (°C)	Chilled Cold-Block (°C)
**Mean ± SE**	**20.76 ± 0.06**	**2.40 ± 0.08**	**21.07 ± 0.06**	**4.98 ± 0.06**

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
