# Peer review of "Aquaporins Respond to Chilling in the Phloem by Altering Protein and mRNA Expression"

_cells, 2019, doi:10.3390/cells8030202_

Round 1
Reviewer 1 Report
The major concern I have is the language and scientific terms used in the article. Many of the sentences are not constructed properly. Apart from this issue, the rest of the MS is excellent and presenting interesting findings.
Abstract
heat exchangers? if possible this can be explained with few words to introduce in the abstract
"aquaporins line the entire source to sink pathway" - need improvement
"These protein water channels show regulatory roles" - better to write directly aquaporin or water transporting AQPs
"PIP2 aquaporin protein signal" - PIP2s expression
Methods
Better to include details of primary antibodies used for immunolocalization.
Results
In qRT-PCR results, no need to highlight fold change where the difference is statistically not significant.
Reviewer 2 Report
Despite great interest of researchers in the study of water channels aquaporins (AQPs), many aspects of their action in plants still remain unclear. The present research demonstrates very fast changes in the level of PIP AQPs present in plasma membranes of Populus balsamifera contributing to release of water from the sieve tubes and normalization of pressure inside phloem after chilling. The effect is novel and important for plant adaptation to cooling and I am sure that it is worth publishing. Still some improvements are needed before accepting the MS.
Major remarks (they are mostly not very great)
1. Line 93. “PIP1 predominantly are found within internal compartments”. Here reference is missing. I am not sure that many reports demonstrated the presence of PIP1 in internal compartments. Its title (Plasma membrane intrinsic proteins) serves as an indicator of its presence in plasma membranes. May be it should better be said that the presence of PIP1 within internal compartments was demonstrated in some cases (it is likely to be reference 52). This remark should be also addressed in the corresponding paragraph of Discussion.
2. Line 177-178 and below (about stomatal conductance). I think that it should be better specified, what kind of changes in stomatal conductance occurred after chilling. The subtraction of stomatal conductance after cold application by pre-cold conductance value and conversion to an absolute value make it unclear, if chilling increased or decreased stomatal conductance.
3. Lines 329-330. “Visually, the area of individual sieve elements appeared to be greatest in the 10 min chill treatment, however no significant difference in area was detected between treatments” – it not clear for me what is meant. Do authors mean that 10 min after chilling the area appeared to be greatest, but this was not statistically verified?
4. Lines 447-448. “We must reject our hypothesis that the localization pattern of aquaporins change in response to cold as we did not find a significant treatment effect of plasma membrane: internally located aquaporins” – This is unclear sentence. Did authors mean distribution of AQPs between plasma membrane and internal compartments?
5. Next sentence. “Since there was a rapid decrease in labeling between the 2 min and 10 min chilling treatments, this begs the question of where the aquaporins went during this time?” – I think it even more important to address the question about the source of the increase in AQPs detected immediately after the chilling. Where this AQPs could have come from? I understand that no final answer can be done, but at least some suggestions are necessary.
Minor remarks
1. Lines 61-62. “The dispersion of these so called forisome p-proteins have been demonstrated to disperse rapidly following cooling due to a depolarization of the sieve element membrane in bean plants” – I think this “dispersion” repeated twice should be somehow changed.
2. Lines 63-64. “However, this explanation does not fully explain why translocation stops in species which do not possess dispersive p-proteins commonly found in legumes [14] or in poplar [19].” – the same concerns “explanation” and “explain” in the next sentence.
3. Lines 71-72. “it is hypothesized that the specific element of the membrane disrupted by cold could be solute and aquaporin water transporters” - I am not sure that the phrase about “solute and aquaporin water transporters” is correct. It sounds as if AQPs are transported
4. Lines 81-80. “The effect of chilling on aquaporins has been shown by a down regulation of mRNA transcript, but an upregulation in protein expression” – it seems to me that the word “corresponding” is missing before “transcript” (not clear, which transcript is meant).
5. Line 422. “water update” – is it really “update” or it should be “uptake”?
Round 2
Reviewer 1 Report
The revised version is perfect, Authors have addressed all of my concern.